# The neural correlates of context driven changes in the emotional response: An fMRI study

**Brigitte Biró**[1,2,3], **Renáta Cserjési**[3], **Natália Kocsel**[3], **Attila Galambos**[2,3], **Kinga Gecse**[1,4], **Lilla Nóra Kovács**[2,3], **Dániel Baksa**[1,4], **Gabriella Juhász**[1,4], **Gyöngyi Kökönyei**[1,3,4]*

1 NAP3.0-SE Neuropsychopharmacology Research Group, Hungarian Brain Research Program, Semmelweis University, Budapest, Hungary, 2 Doctoral School of Psychology, ELTE Eötvös Loránd University, Budapest, Hungary, 3 Institute of Psychology, ELTE Eötvös Loránd University, Budapest, Hungary, 4 Faculty of Pharmacy, Department of Pharmacodynamics, Semmelweis University, Budapest, Hungary

* kokonyei.gyongyi@pharma.semmelweis-univ.hu, kokonyei.gyongyi@ppk.elte.hu

**Data Availability Statement:** All data underlying the findings described in the manuscript, i.e., post-scan valence and arousal ratings of the picture pairs and reaction times to the second pictures in the MR scanner along with main fMRI contrast

## Abstract

Emotional flexibility reflects the ability to adjust the emotional response to the changing environmental context. To understand how context can trigger a change in emotional response, i.e., how it can upregulate the initial emotional response or trigger a shift in the valence of emotional response, we used a task consisting of picture pairs during functional magnetic resonance imaging sessions. In each pair, the first picture was a smaller detail (a decontextualized photograph depicting emotions using primarily facial and postural expressions) from the second (contextualized) picture, and the neural response to a decontextualized picture was compared with the same picture in a context. Thirty-one healthy participants (18 females; mean age: 24.44 ± 3.4) were involved in the study. In general, context (vs. pictures without context) increased activation in areas involved in facial emotional processing (e.g., middle temporal gyrus, fusiform gyrus, and temporal pole) and affective mentalizing (e.g., precuneus, temporoparietal junction). After excluding the general effect of context by using an exclusive mask with activation to context vs. no-context, the automatic shift from positive to negative valence induced by the context was associated with increased activation in the thalamus, caudate, medial frontal gyrus and lateral orbitofrontal cortex. When the meaning changed from negative to positive, it resulted in a less widespread activation pattern, mainly in the precuneus, middle temporal gyrus, and occipital lobe. Providing context cues to facial information recruited brain areas that induced changes in the emotional responses and interpretation of the emotional situations automatically to support emotional flexibility.

## Introduction

Emotional flexibility refers to the ability to modulate one's emotional responses to fit the changing demands of the environmental context, and, thus, to change–generate, inhibit, down- or upregulate–one's initial emotional responses according to the contextual demands

maps, are fully available at https://osf.io/hgdky/. However, we are not allowed to share raw imaging dataset publicly, because at the time our study started, there was no information on open access data availability in the consent forms (the study was approved by the Scientific and Research Ethics Committee of the Medical Research Council (Hungary)), therefore participants were not able to accept or refuse their assent to share imaging data in an open access repository. However, raw imaging data are available from the corresponding author (Gyöngyi Kökönyei, kokonyei.gyongyi@ppk.elte.hu) or from the Department of Pharmacodynamics, Faculty of Pharmacy, Semmelweis University (titkarsag.gyhat@pharma.semmelweis-univ.hu) on reasonable request.

**Funding:** This study was supported by the Hungarian Academy of Sciences (MTA-SE Neuropsychopharmacology and Neurochemistry Research Group), the Hungarian Brain Research Program (Grant: 2017-1.2.1-NKP-2017-00002), and the Hungarian Brain Research Program 3.0 (NAP2022-I-4/2022), and by the Hungarian National Research, Development and Innovation Office (Grant No. FK128614, K 143391). Project no. TKP2021-EGA-25 has been implemented with the support provided by the Ministry of Innovation and Technology of Hungary from the Hungarian National Research, Development and Innovation Fund, financed under the TKP2021-EGA funding scheme. DB was supported by the ÚNKP-20-3-II-SE-51 New National Excellence Program of the Ministry for Innovation and Technology from the source of the National Research, Development and Innovation Fund. The sponsors had no role in the design of study, in the collection, analysis, interpretation of data and in the writing the manuscript.

**Competing interests:** The authors have declared that no competing interests exist.

[1,2]. Context can automatically direct emotional processing and can easily override facial expressions [3]. Learned associations between emotional responses and contexts lead to the appraisal of situations and shape the emotional responses and/or the regulatory processes [4,5]. For illustration, crying at a funeral represents sadness, in contrast with crying when achieving great success it represents happiness or pride. Thus, the meaning of a stimulus may depend on the context and may change when the context changes, so one of the key elements of emotional flexibility is shifting between meanings to adapt our behavior, e.g., our emotional response to the context. In short, to give an appropriate emotional response after context modifications [6,7].

Decoding emotions from faces has been extensively studied, but there is now a large body of evidence proving that the situational context (e.g., physical and social environment), along with the emotional/social knowledge of the perceiver about the situation will automatically guide the perception [8], causing even radical categorical changes in the perceived emotion (e.g., pride instead of sadness) [9]. Contextual information is processed and integrated with facial affective information in the early phase of perception [10]; thus, even the perception of a basic facial emotion can be categorically changed automatically by the context [9].

There are many ways to test the effect of context on emotion perception experimentally. For instance, knowledge about the situation of the observed person can be manipulated by semantic-linguistic labels [3,11] and information given before presenting even neutral faces [12]. Emotional faces can also be presented on different, even artificial backgrounds [13,14], or in naturalistic scenes. Results of electrophysiological [10] and magneto-encephalographic [15] studies suggest that facial perception is influenced by contextual cues even in the early stage of visual processing. On the basis of the available evidence, Aviezer and colleagues [16] conclude that context does not simply have a modulating effect on the processing and perception of emotions, but can actually lead to a categorical shift in the perception of an emotional expression.

Research on cognitive reappraisal can also help to understand the impact of context on emotional information processing. In reappraisal studies [17,18], when participants are asked to give a different meaning to a negative or positive stimulus, they are instructed to create a new cognitive context for the stimulus. Shifting from emotional to non-emotional or from negative to positive meaning (or vice versa) definitely alters the emotional trajectory, causing changes in the initial emotional responses [19]. In a multi-level framework, proposed by Braunstein, Gross, and Ochsner [20] reappraisal studies are considered to address controlled emotion regulation with explicit regulatory goals: participants are instructed to regulate their emotions (explicit regulatory goal) using effortful processes to change the cognitive context (controlled processes). However, reappraisal or shift in meaning, or more generally, emotion regulation can happen without explicit regulatory goals and/or in a more automatic manner (see [20,21]).

To extend previous works, our aim was to investigate a shift or change in emotional perception triggered by the context that occurs automatically and without explicit instructions to change. We used the Emotional Shifting Task (EST) [22,23], in which pairs of pictures are presented: the first one is a small detail of the second picture, as depicted in Fig 1. The presentation of the first picture, which is a decontextualized part of the whole picture, generates an emotional response, but when this picture is put into a context, the context itself may cause a change (and in some cases a shift) in the meaning and valence of the stimulus. For instance, a picture of a smiling girl is generally evaluated as a positive stimulus but if it turns out that she is smiling while bullying a peer, it will probably cause a shift in the evaluation toward a negative direction automatically and without any explicit regulatory goals. This technique was developed first by Munn in 1940 [24], who selected 16 pictures depicting emotional

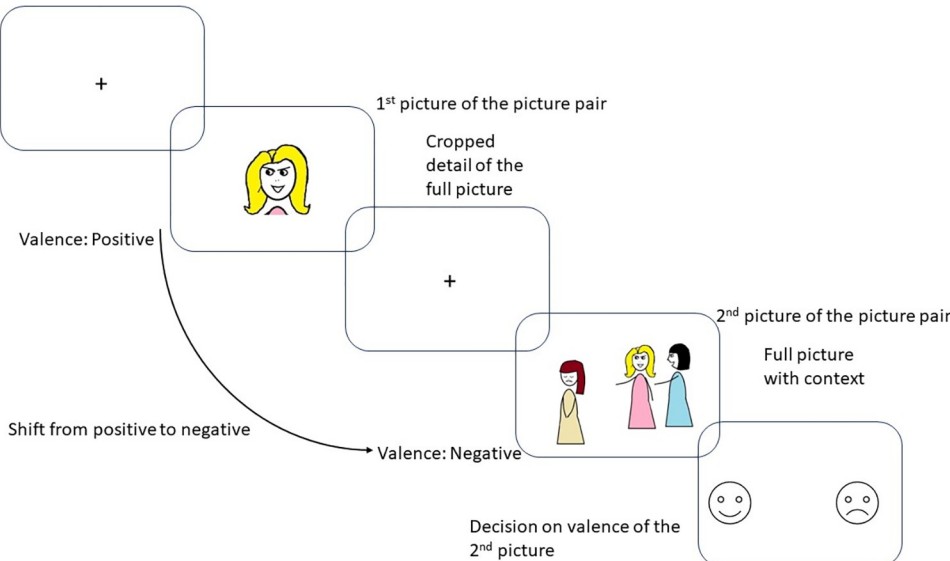

**Fig 1. Schematic representation of a sequence of trial in the Emotional Shifting Task.** An example for shifting from positive to negative emotion.

expressions from Life and Look magazines and then prepared two sets of pictures: one set with the full picture including the context as well, another set with only the face from the full picture. Munn found that the extra information of the context could change the judgment of an emotional face when participants were asked to name the emotion appearing on the face. The EST also uses naturalistic scenes as well in order to mimic real-life emotion perception and increase the ecological validity of our task.

Our aim was to explore the neuronal responses to the previously seen decontextualized photograph (depicting emotions using primarily facial and postural expressions) when it was presented in a naturalistic context. We expected that areas involved in processing of facial and contextual cues were going to be recruited by our task. For instance, studies using facial expressions in investigating emotion perception found that facial expressions activated the so-called face-selective regions, including the inferior occipital gyrus (occipital face area, OFA), lateral fusiform gyrus (fusiform face area), and posterior superior temporal sulcus (pSTS) as core regions of a widely distributed network [8]. Other brain regions such as the amygdala, anterior inferior temporal cortex [25], insula, and inferior frontal gyrus [26] also play a role in facial emotional processing [27]. As for the context, it is well established that complex social situations can easily trigger mentalizing, i.e., inferring mental and affective states to others (often called theory of mind) [28,29] and/or empathic responding, i.e., vicariously experiencing the feelings and emotional states of other people [30]. Core brain regions of the neural network of mentalization are the bilateral temporoparietal junction (TPJ) and medial prefrontal cortex (mPFC) [31] whereas the core network of empathy involves the dorsal anterior cingulate cortex, anterior midcingulate cortex and supplementary motor area (SMA) [32].

The EST task we used allowed us to distinguish different conditions as presented in Table 1. On the basis of two dimensions, which are (1) the valence (positive or negative) of the initial emotional stimulus and (2) the valence (positive or negative) of its contextualized presentation, four different types of automatic changes can be targeted in emotional response. We argue that emotional flexibility can be investigated with this design as Coifman and Summers [2] pointed out the initial emotional response (to the decontextualized pictures) must be modulated (upregulate the emotions or shift) to fit to the context.

**Table 1. Understanding emotional flexibility in terms of context-driven emotional response.**

|  |  | Second picture (with context) | |
|---|---|---|---|
|  |  | Positive | Negative |
| **First picture** (no context) | **Positive** | Upregulation of positive emotions | Shift from positive emotions to negative ones |
|  | **Negative** | Shift from negative emotions to positive ones | Upregulation of negative emotions |

By shift we refer to categorical changes where the valence of the initial emotional response is reversed, i.e., the initial negative emotional response becomes positive and vice versa. The term upregulation is used when the valence of the same initial emotional response is increased by the context; thus, a negative stimulus becomes more negative, or a positive one becomes more positive. Accordingly, the EST contained two shift and two non-shift (upregulation) conditions.

On the basis of the theory by Saxe and Houlihan [33] different emotional responses could be expected to stimuli in context vs. without context. They argue that forward inferences are used to attribute emotions to the target when an emotional expression is processed in a context; thus, we automatically infer that the cause of the emotional state of the target reflected in their emotional expressive behavior is the context/event. On the basis of this, we expected that context itself would recruit areas involved in emotional processing and understanding complex social situations; thus, first we simply compared the neural responses to whole pictures vs. decontextualized (cropped) pictures. We refer to this as a general context effect in our study. Then we used this activation map as an exclusive mask to be able to explore the four different types of automatic changes specifically in emotional responses. It allowed us to explore neural activation to changes in the meaning triggered by the context as a passive cue independent of the context vs. no context differences. On the basis of previous studies, we hypothesized that prefrontal regions, especially the mPFC and dorsolateral prefrontal cortex (dlPFC) [17,34] were going to be recruited when the context induced a shift in the emotional valence of the pictures. More specifically, on the basis of a recent study on the automatic regulation of negative emotions by Yang and coworkers [35] we expected that visual areas, striatal areas, precentral/postcentral gyri and dlPFC would be activated when context resulted in a shift from negative towards positive meaning.

## Method

### Participants

Thirty-two healthy adult volunteers recruited through social media sites and journal advertisements were included in the present study; however, one participant was excluded from the first level analysis due to excessive movement during the fMRI measure; thus, the final data of 31 participants, 18 females and 13 males (mean age: 24.44 ± 3.4), were analyzed. The participants were right-handed, as assessed by the Edinburgh Handedness Inventory [36], and had normal or corrected-to-normal vision. All participants were examined by a senior psychiatrist and neurologist and were excluded with any history of psychiatric or neurological disorders or chronic medical conditions.

The present study was approved by the Scientific and Research Ethics Committee of the Medical Research Council (Hungary), and written informed consent was received from all subjects in accordance with the Declaration of Helsinki.

### Psychological task: The Emotional Shifting Task

The EST [22] consists of 24 picture pairs. In each pair, the first picture is always a smaller detail from the second (whole) picture. In most cases the cropped image expressed emotion

**Fig 2. The design of the Emotional Shifting Task.**

primarily through facial expression and/or posture. The valence of the firstly presented picture either remains or changes when it is placed into a context, and so should change the elicited emotion (Fig 1). For the upregulation conditions (P1P2 and N1N2), pictures were selected from the International Affective Picture System [37]. Their identification numbers were 1340, 2091, 2141, 2205, 2216, 2340, 2530, 2700, 6242, 6838, 8497, and 9050. For the shift conditions (P1N2 and N1P2) pictures were selected from the internet. Six criteria were used to select the images: (1) free for non-commercial use, (2) depicting social interactions, (3) evoking an emotional response without being shocking or extreme, (4) not depicting famous person(s), (5) eligible for shifting conditions, i.e., the valence of facial expression and the whole picture should be opposite, and (6) the images should represent as many different situations as possible.

After each pair, a happy and a sad smiley/emoji appeared on the screen (Fig 2), and participants had to choose one of them by pressing the corresponding button to indicate the valence (positive or negative) of the second (whole) picture. We decided to use emojis in the scanner to mimic the two endpoints of valence ratings in Self-Assessment Manikin [37].

Four conditions were defined in the task: two conditions, in which participants were required to alter their emotions either from positive to negative (P1N2) or from negative to positive (N1P2), and two conditions, where no shift (but upregulation) was expected in the valence (i.e. both pictures presented were either positive (P1P2) or negative (N1N2)). Each condition consisted of six pairs of pictures, presented in pseudo-random order. During functional magnetic imaging, two behavioral variables were registered: the reaction times (RT) of the selection of the emoji and the number of "correct" answers. An answer was considered correct if the valence of the secondly presented picture matched the valence of the selected smiley. Stimulus presentations and data registrations were conducted in E-Prime 2.0 Software (Psychology Software Tools, Inc., Pittsburgh, PA, USA).

## Procedure

Data collection included three steps. First, participants completed a short practice session, which was explained and presented on a laptop outside of the scanner and consisted of three pairs of stimuli that were not used in the task and included shift and non-shift conditions as well. In this part, participants could read the instructions and ask their questions in case of uncertainty regarding the instructions or the operation of the task.

In the next step, participants were instructed to get as emotionally involved in the presented situations as possible while viewing the pictures in the scanner. To measure a baseline brain activity, a white fixation cross was presented on a black background at the beginning and at the end of the task for 20 seconds. Each emotional stimulus was shown for 8 seconds. The timing was based on laboratory pilot studies and previous studies [11]. To avoid artifacts due to expectations, fixation crosses were presented with altered timing (from 5 to 11 seconds, mean presentation time: 8 seconds) before each emotional stimulus. At the end of each trial, the answer screen was presented for 4 seconds (Fig 2).

Lastly, a post-test was filled out after the fMRI measure (outside the scanner) to examine whether participants observed changes in the valence of the pictures. In this part, the pairs of pictures were presented in the same order on a laptop, and participants were asked to rate them on a 7-point Likert scale. Valence and arousal were measured from 1 to 7 (1 being very unpleasant and 7 very pleasant; 1 being calm and 7 very excited, respectively).

## FMRI acquisition

The functional MRI data collection was carried out by a SIEMENS MAGNETOM Prisma syngo MR D13D 20 channels headcoil 3T scanner. A BOLD-sensitive T2*-weighted echo-planar imaging sequence was used (TR = 2220ms, TE = 30 ms, FOV: 222) with 3 mm × 3 mm in-plane resolution and contiguous 3-mm slices providing whole-brain coverage. Four hundred and nine volumes were acquired during the task. For the structural data a series of high-resolution anatomical images were acquired before the functional imaging using a T1-weighted 3D TFE sequence with $1 \times 1 \times 1$ mm resolution.

## Statistical analysis of self-report and post-test data

To analyze demographic and behavioral data SPSS version 28.0 (IBM SPSS, IBM Corp, Armonk, NY, USA) was used, and descriptive and non-parametric statistics were performed. As the distribution of valence and arousal ratings was non-normal, we used Wilcoxon Signed Rank Test to compare the valence and arousal ratings of the first and second pictures in each condition (P1N2, N1P2, P1P2, and N1N2). However, as it was easier to interpret changes in means than in ranks, we repeated these analyses using a series of bootstrapped paired t-tests. A repeated measures ANOVA was performed on the reaction times collected during the fMRI scan.

## FMRI data analyses

**Preprocessing.** Statistical Parametrical Mapping (SPM12) analysis software package (Wellcome Department of Imaging Neuroscience, Institute of Neurology, London, UK; http://www.fil.ion.ucl.ac.uk/spm12/ implemented in Matlab 2016b (Math Works, Natick, MA, USA) was used to analyze the imaging data. Preprocessing contained the following steps: realignment, co-registration to the structural image, segmentation, normalization in the Montreal Neurological Institute (MNI) space, and spatial smoothing with an 8-mm full-width half-maximum Gaussian kernel. These steps of preprocessing were performed on the functional images. Finally, a visual inspection of the pictures took place to exclude the poor-quality images.

**First level model.** During first-level analyses, BOLD (blood oxygenation level-dependent) hemodynamic responses were modeled in a general linear model. In the event-related single subject analysis fixation screens, both stimuli (positive and negative), the disposition of the shift (the valence of the first stimuli: positive/negative, and the nature of the condition: shift/non-shift condition) and the two possible answers (happy and sad emojis) were modeled as separate regressors of interest. High-pass temporal filtering with a cut-off of 128 s was included in the model to remove the effects of low-frequency physiological noise, and serial correlations in data series were estimated using an autoregressive AR (1) model. Motion outliers (threshold of global signal > 3 SD and motion > 1 mm) were identified with the Artifact Detection Tools (ART; www.nitrc.org/projects/artifact_detect/), and the six motion parameters were used as regressors of no interest in the fMRI model.

Four contrasts were created to analyze whether an increased activation could be detected to stimuli that were placed into a context (2nd picture) compared to the ones without contextual background (1st picture), and also focusing on the valence of the stimuli (see Table 1).

**Second-level analyses.** During second-level analyses (whole brain t-test) the threshold was set to p< .05 family-wise error (FWE) corrected for multiple comparisons. The automated anatomical labeling atlas (aal) [38] was used to anatomically identify the activated clusters, whereas the MNI 152 template brain provided in MRIcroGL was used to visualize statistical maps http://www.mccauslandcenter.sc.edu/mricrogl/ [39].

## Results

### Behavioral results

**Descriptive statistics.** To track the changes in valence and arousal in picture pairs, ratings of the valence and arousal values registered in the post-task after scan were analyzed (S1 Table). Answers were compared to the post-task, and the valence ratings of all the pictures were in the expected directions, namely positive pictures (P1; P2) were rated more pleasant and negative pictures (N1; N2) more unpleasant. Results of the Wilcoxon Signed Rank Test (and bootstrapped paired t-test) showed the significant differences between the mean valence and arousal values within conditions according to which valence ratings significantly differed in each condition (P1N2; P1P2; N1P2; N1N2), similarly, arousal ratings showed a significant increase for the second pictures (P2, N2), compared to the first ones (P1, N1) in each condition (see S1 Table, Figs 3 and 4).

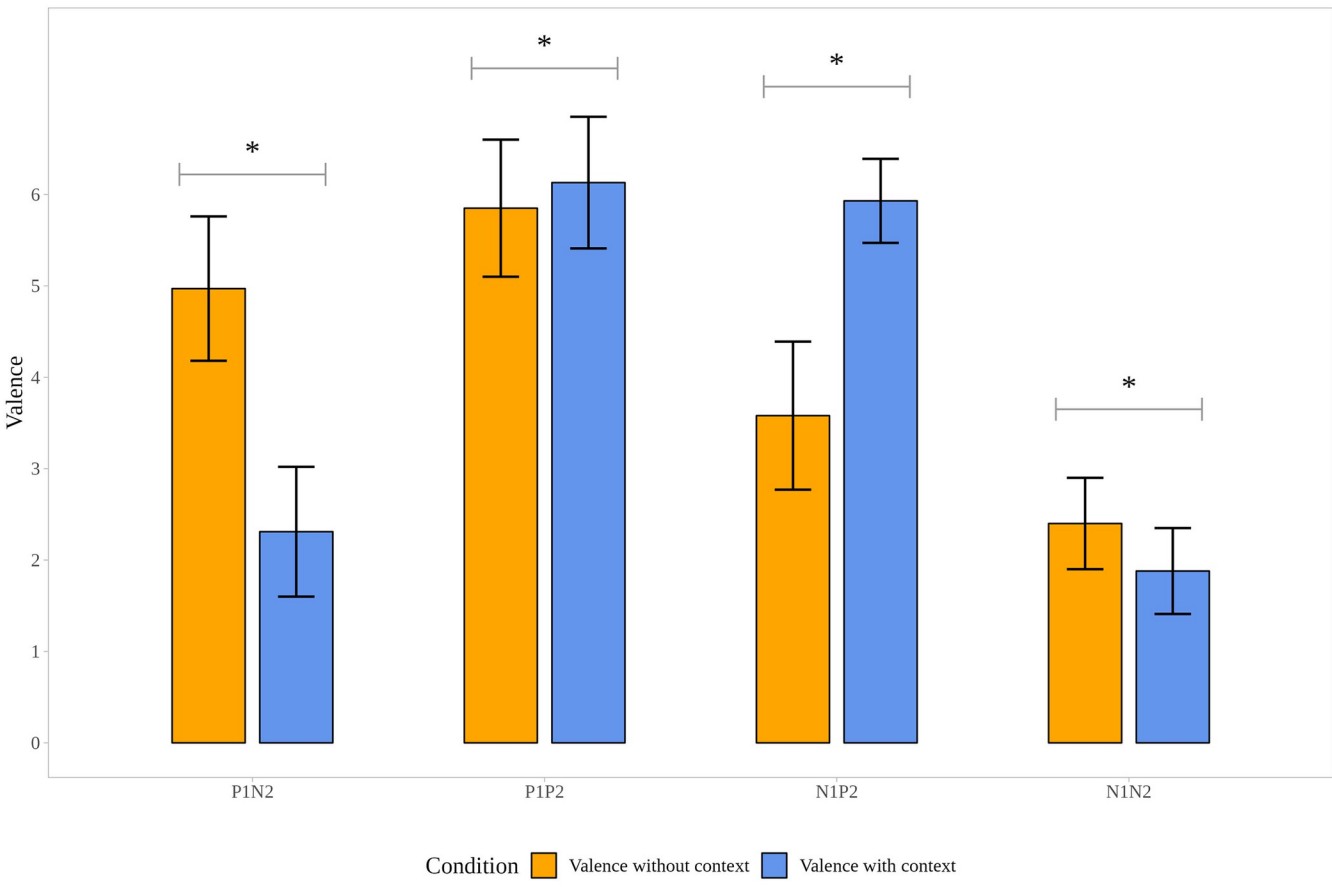

**Fig 3. Changes in valence ratings in the post-task.** Note. P1: First picture of the picture pairs is positive. N1: First picture of the picture pairs is negative. P2: Second picture of the picture pairs is positive. N2: Second picture of the picture pairs is negative. * p < .001.

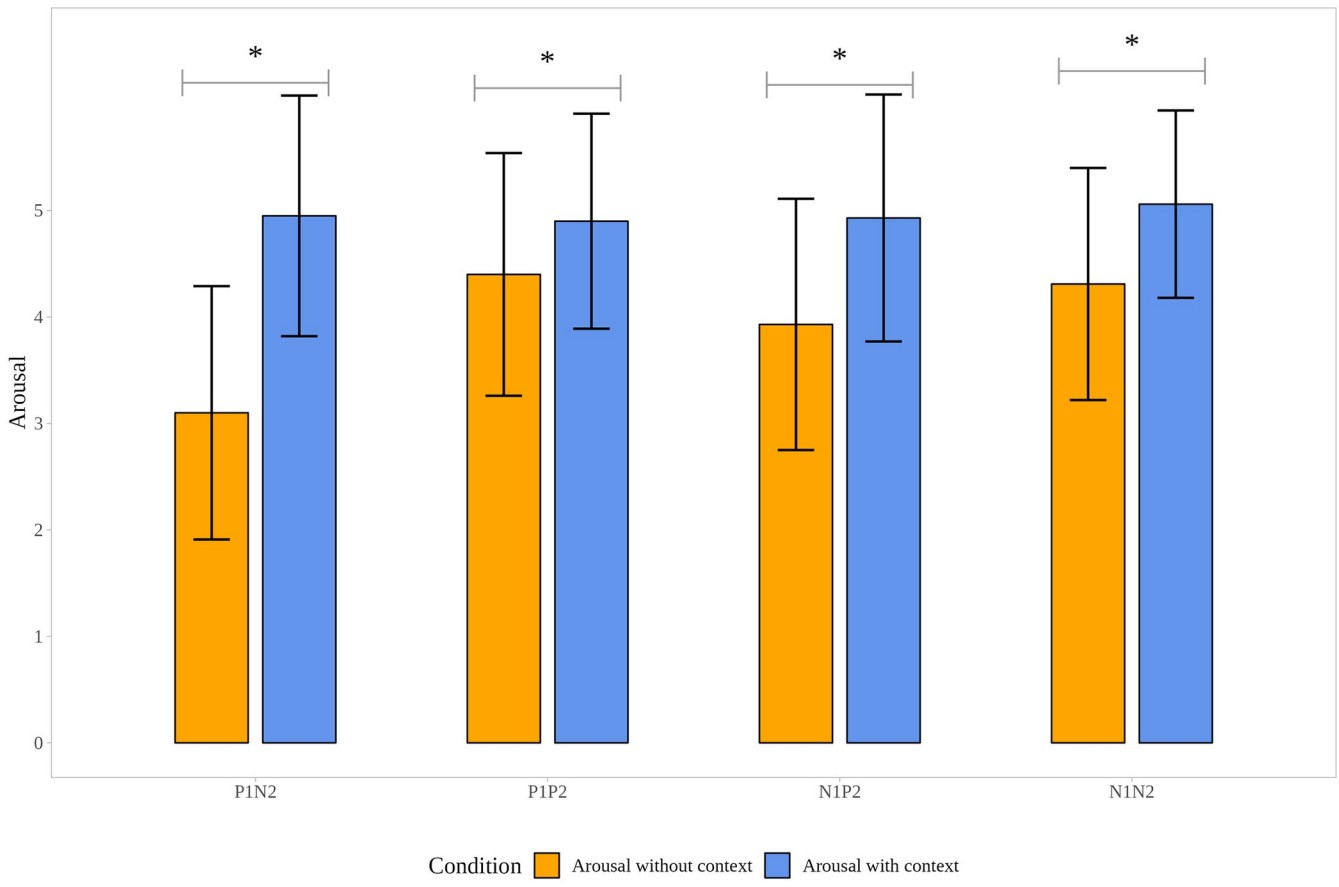

**Fig 4. Changes in arousal ratings in the post-task.** Note. P1: First picture of the picture pairs is positive. N1: First picture of the picture pairs is negative. P2: Second picture of the picture pairs is positive. N2: Second picture of the picture pairs is negative. * p < .001.

Descriptive data, collected during the EST, provided information on the mean reaction time values given in the different contrasts (see S2 Table) and also on the "accuracy" of the answers of the participants (whether they picked the expected smiley/emoji showed on the screen). This accuracy was 95.54% (range: 79–100%). A repeated measures ANOVA was performed on the reaction times collected during the fMRI scan, resulting in a significant difference across the four contrasts ($F_{3, 90} = 16.273$, p< .001). Post hoc analyses showed that subjects pressed the button more slowly in trials of shifting from positive to negative (P1N2) compared to the other three types of trials. Reaction times in the trials when both pictures were negative (N1N2) were longer, compared to the trials when both pictures were positive (P1P2) and to the trials of shifting from negative to positive (N1P2) (see S2 Table and Fig 5).

## Task-related activations

**Main effect of context.** The main effect of the context was checked by comparing the increased brain activations of the second images (full pictures with the context) to the firstly presented images (pictures without context) regardless of valence changes. Widespread activations were found in the brainstem, lingual and fusiform gyri, precuneus, calcarine, middle and superior temporal gyri, middle and superior occipital gyri, inferior parietal gyrus, middle, superior, medial, inferior frontal gyri, precentral gyrus, SMA, anterior cingulate (ACC) and postcentral gyrus (S3 Table and S1 Fig).

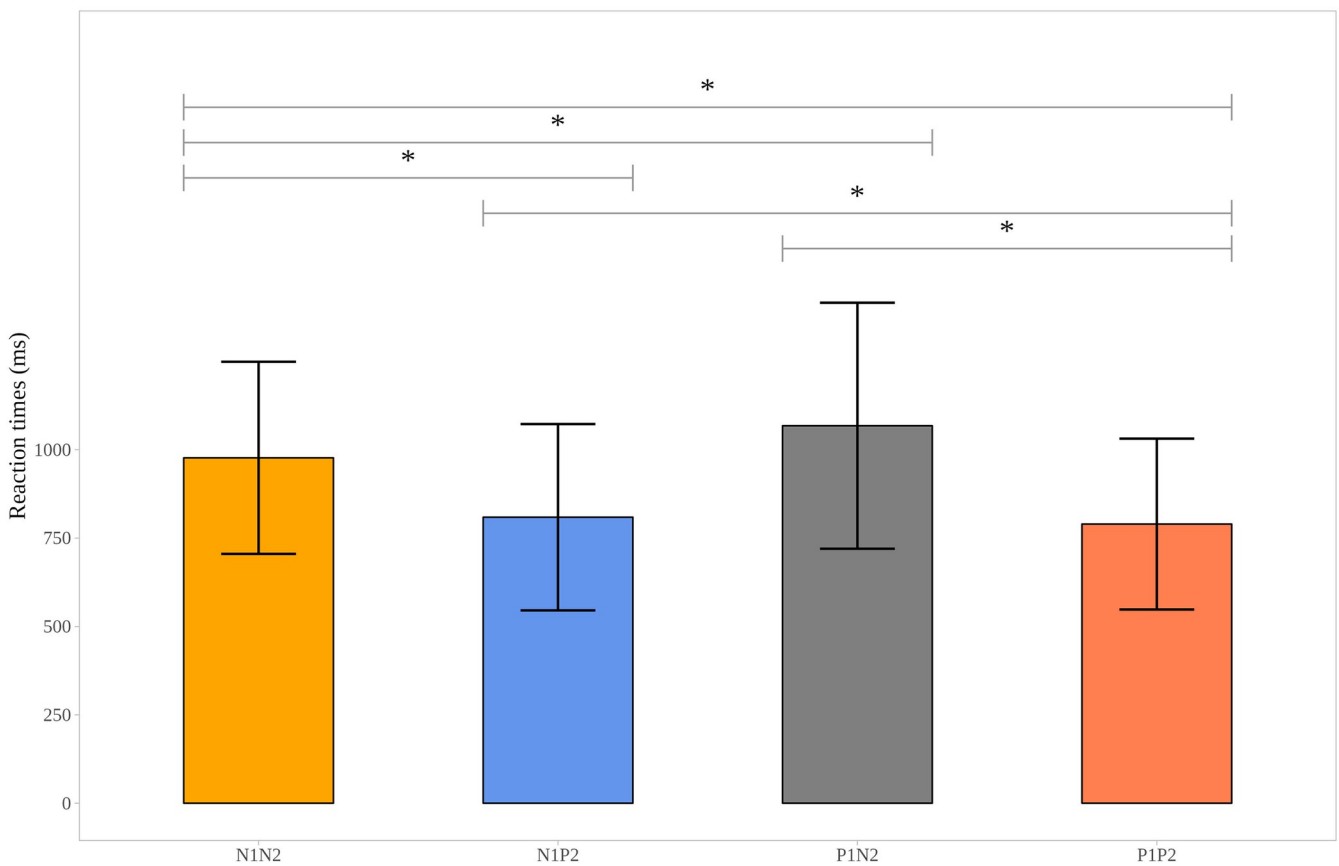

**Fig 5. Mean reaction times (and standard deviations in milliseconds) to the second picture of the picture pairs in the scanner by the type of picture pairs in the Emotional Shifting Task.** Note. P1: First picture of the picture pairs is positive. N1: First picture of the picture pairs is negative. P2: Second picture of the picture pairs is positive. N2: Second picture of the picture pairs is negative. * p < .05.

**From positive to negative: Activation to positive pictures in a negative context compared to positive pictures without context (P1-N2).** To reveal the increased brain activation specific to the change in the meaning from positive to negative triggered by context, the context main effect regions were used as an exclusive mask on the results of the widespread increased activations to positive pictures in a negative context compared to positive pictures without context (Table 2 and Fig 6). Thus, the regions activated outside the mask were the superior medial frontal gyrus, inferior orbitofrontal gyrus, superior temporal pole, middle and superior temporal gyrus, middle occipital gyrus, SMA, anterior cingulum, thalamus, caudate and amygdala.

**From negative to positive: Brain activations to negative pictures in a positive context compared to negative pictures without context (N1P2).** In picture pairs, where the secondly presented positively valenced stimuli (P2) were compared to the firstly presented negatively valenced stimuli (N1), increased activations were found in the middle temporal gyrus, middle cingulum, middle occipital gyrus, precuneus and calcarine when we used the above mentioned exclusive mask (Table 2 and Fig 6).

**From positive to positive: Brain activations when both the first and second pictures were positive (P1P2).** Increased BOLD signals were found in the visual areas including the

**Table 2. Brain regions showing increased activation to positive pictures in a negative context compared to positive pictures without context and brain regions showing significantly increased activation to negative pictures in a positive context compared to negative pictures without context.**

| Cluster size (voxel) | Region | Side | Peak T-values | MNI coordinates | | |
|---|---|---|---|---|---|---|
| | | | | x | y | z |
| **Positive pictures in a negative context > Positive pictures without context** | | | | | | |
| 228 | Superior Frontal Gyrus, medial | L | 9.03 | 0 | 56 | 23 |
| | Pregenual ACC | R | 7.95 | 6 | 50 | 17 |
| | Superior Frontal Gyrus, medial | R | 7.94 | 6 | 47 | 26 |
| 40 | Inferior frontal Gyrus, pars orbitalis | L | 7.65 | -45 | 23 | -10 |
| | Superior Temporal Pole | L | 7.04 | -36 | 17 | -25 |
| | Superior Temporal Pole | L | 6.80 | -42 | 17 | -19 |
| 14 | Thalamus | R | 7.54 | 9 | -25 | -1 |
| 66 | Caudate | R | 7.35 | 12 | 11 | 8 |
| | Caudate | R | 7.19 | 6 | 8 | -1 |
| | Thalamus | R | 6.54 | 6 | -7 | 8 |
| 12 | Middle Occipital Gyrus | L | 7.24 | -39 | -76 | 11 |
| 17 | Rectus | L | 7.10 | 0 | 50 | -16 |
| | Superior Frontal Gyrus, medial orbital | L | 6.31 | 0 | 59 | -13 |
| 26 | Thalamus | L | 6.78 | -6 | -7 | 5 |
| | Caudate | L | 6.56 | -9 | 8 | 5 |
| | Caudate | L | 6.10 | -12 | 14 | -1 |
| 2 | Amygdala | L | 6.68 | -21 | -7 | -16 |
| 48 | Supplementary Motor Area | R | 6.66 | 6 | 17 | 68 |
| | Superior Frontal Gyrus, medial | R | 6.59 | 15 | 32 | 59 |
| | Supplementary Motor Area | R | 6.42 | 3 | 14 | 56 |
| 14 | Middle Temporal Gyrus | L | 6.37 | -60 | -10 | -16 |
| 12 | Middle Temporal Gyrus | R | 6.37 | 57 | -37 | -1 |
| | Superior Temporal Gyrus | R | 6.18 | 48 | -37 | 5 |
| **Negative pictures in a positive context > negative pictures without context** | | | | | | |
| 71 | Middle Temporal gyrus | R | 9.05 | 51 | -73 | 2 |
| | Middle Temporal gyrus | R | 8.08 | 39 | -67 | 20 |
| | Middle Temporal gyrus | R | 7.31 | 42 | -73 | 8 |
| 45 | Superior Occipital Gyrus | R | 7.75 | 33 | -70 | 41 |
| 18 | Precuneus | R | 7.40 | 3 | -52 | 56 |
| | Precuneus | R | 6.17 | 6 | -64 | 47 |
| 25 | Calcarine | R | 7.40 | 12 | -91 | 8 |
| | Calcarine | R | 6.88 | 18 | -88 | -1 |
| 11 | Middle Occipital Gyrus | L | 7.22 | -39 | -79 | 2 |
| 16 | Midcingulate | R | 6.17 | 6 | -52 | 32 |
| | Precuneus | | 6.06 | 0 | -55 | 20 |
| | Precuneus | R | 5.93 | 3 | -61 | 29 |

Note. R = right, L = left; the statistical threshold was set to p< .05, family-wise error (FWE) corrected for multiple comparison, ACC = anterior cingulate cortex.

calcarine and the lingual gyrus (Table 3) when positive pictures in a positive context were compared to positive pictures without context when using the exclusive mask.

**From negative to negative: Brain activations when both the first and second pictures were negative (N1N2).** The superior occipital gyrus showed an increased activation (Table 3) to negative pictures in a negative context compared to negative pictures without context, and the context main effect regions were used as an exclusive mask.

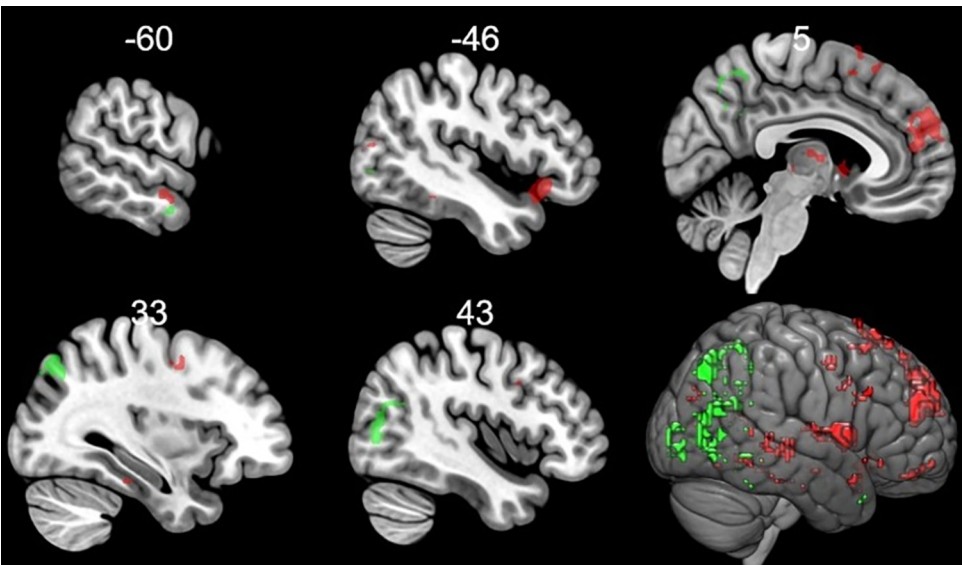

**Fig 6. Activated regions when the context categorically changed the valence of the emotional stimuli (from negative to positive and from positive to negative) after excluding the general effect of context.** Increased activations when a negative stimulus was put in a positive context are shown in green whereas increased activations when a positive stimulus was put in a negative context are shown in red. Coordinates are in the Montreal Neurological Institute (MNI) space. Statistical maps were visualized on the MNI 152 template brain provided in MRIcroGL [39].

## Discussion

In the present study we aimed to measure emotional flexibility, defined here as a change in the emotional response elicited by a specific context. This task is built on the notion that the context may give an entirely different interpretation to the stimulus, resulting in even categorical changes in the valence and/or arousal of the elicited emotion [2]. In our task, the context appeared as a passive cue that induced changes in the emotional responses and interpretation of the emotional situations automatically; thus, this task is considered to explore changes in the spontaneous emotional output guided by the context. Thus, on the basis of the review by Coifman and Summers [2], we argue that EST is an appropriate task to capture emotional

**Table 3. Brain regions showing significantly increased activation to positive pictures in a positive context were compared to positive pictures without context (P1P2 picture pairs), and those with significantly increased activation to negative pictures in a negative context were compared to negative pictures without context (N1N2 picture pairs).**

| Cluster size (voxel) | Region | Side | Peak T-values | MNI coordinates | | |
|---|---|---|---|---|---|---|
| | | | | x | y | z |
| Negative pictures in a negative context > negative pictures without context | | | | | | |
| 15 | Superior Occipital Gyrus | R | 7.42 | 27 | -73 | 44 |
| Positive pictures in a positive context > positive pictures without context | | | | | | |
| 17 | Lingual Gyrus | R | 7.24 | 12 | -88 | -4 |
| | Calcarine | R | 7.23 | 15 | -94 | 2 |
| 13 | Calcarine | L | 7.18 | -6 | -94 | 2 |
| | | | | -3 | -85 | -4 |

Note. R = right, L = left; the statistical threshold was set to p< .05, family-wise error (FWE) corrected for multiple comparison.

flexibility. Post-task data on the valence, arousal and changes in BOLD responses support that this task can induce changes in the emotional response.

## Context effect on valence, arousal rating and reaction times

To understand the outcome of the fMRI results better, valence and arousal ratings of the emotional stimuli were collected during the post-task (out of scanner post-task): participants were asked to rate the pictures used in the fMRI task after the scan. Significant changes in the valence and arousal ratings of the stimuli were observed after they were placed into a context, indicating that participants reinterpreted the emotional stimuli. These changes in the valence and arousal were detected in all four types of picture sets, indicating that the context itself did not only shape the categorization of the emotional states [9], but it also might affect valence and intensity (arousal). The context, or more precisely the appraisal of overall context (e.g. bullying, being in a hospital with a sick person, or childbirth), i.e., the semantic features, might provide extra affective information, as it gives an explanation for the emotion; thus, it guides our interpretation [33]. For instance, seeing a crying woman in hospital compared to seeing just a crying woman, could elicit a more intense emotional response, as it can activate additional emotional meaning or knowledge such as her relative being sick.

Reaction times in the scanner were also registered and showed that participants performed more slowly to the second picture in trials of shifting from positive to negative compared to the other three types of trials. Reaction times to the second picture were longer in the trials when both pictures were negative compared to trials when both pictures were positive or when shifting from negative to positive was required. Thus, reaction time was longer when the second picture was negative compared to when it was positive. We did not ask our participants to choose an emoji after the first picture in the scanner; thus, it limits our understanding.

This result is in line with the findings of Sakaki and coworkers [40] who presented negative, neutral and positive pictures in their study and found that participants had longer reaction times to negative pictures compared to neutral or positive pictures in their task that required semantic processing. On the basis of the results, Sakaki et al. [40] concluded that facing pictures of negative emotional events affected or interfered with the semantic processing of the following stimuli even more than perceptual processing.

## General context effect on brain activation

Overall, we found increased occipital cortex activation when emotional stimuli were put into context, indicating heightened perception and attention [41] presumably evoked by the information about the social and physical environment surrounding the expressor, along with existing emotional knowledge about the context. The activation of the lateral occipital cortex has been found in studies investigating emotional scene processing [42], and the role of calcarine in visual information processing [43,44] and in visual-imagery processes is well-documented [45].

In our task context brought new knowledge (information) on the emotional state of the protagonist. Indeed, many of the activated areas such as the middle temporal gyrus (MTG), fusiform gyrus, SMA [34], temporal pole [46] caudate [47], brainstem [48] and thalamus [49,50] suggest that the social and emotional meaning of facial/postural information, possibly along with other faces on the picture, required an increased emotional reprocessing when the context appeared.

The mental states we attribute to others and the extent to which we resonate with their emotional states can guide our behavior in complex social situations. Accordingly, the contextual presentation of emotional stimuli may activate brain areas involved in mentalization and/or

empathic response. Thorough investigation of activation maps revealed that the MTG activation extended to the TPJ. Several meta-analyses suggest [31,51,52] that TPJ is one of the core regions associated with social cognition, or more specifically mentalizing (often called theory of mind) [28,53], and TPJ is also activated in empathy studies [51]. Furthermore, the precuneus was also activated, this area was previously associated with affective mentalizing [54] i.e., inferring affective states of others, occasionally correspondingly called cognitive empathy. The recruitment of TPJ and the precuneus in our task might suggest that when context was added to the first picture the observers (our participants) reflected on the emotional meaning of the situation.

We detected increased anterior insula activation when context was added to the first picture. Its role in processing interoceptive information that is a key to representing emotional experience [55] is well-established, and insula activation correlated with self-reported arousal [56]. In a recent study [57] activations in other regions, such as in the superior temporal sulcus, fusiform gyrus and lateral occipital cortex, have also been associated with arousal. Note that our post-test after scanning revealed that the second (whole) pictures were more arousing compared to the first one.

We also found increased activation in the right inferior frontal gyrus and lateral orbitofrontal regions when context was added to the first picture. According to a meta-analysis [58] major overlaps in both regions along with the insula and temporal structures can be seen in emotional processing, interoceptive signaling, and social cognition, supporting previous neuroimaging data and our previous expectations that adding a context will recruit areas involved in emotional processing and understanding complex social situations. These results indirectly support the theory of Saxe and Houlihan [33] proposing that information about the event in which the target is expressing emotion is used as a cause for inferring target's emotion(s).

## Specific context effect

**Placing positive stimulus in negative context.**   After excluding the general effect of context by using a mask with activation to context vs. no-context, the automatic shift from positive to negative valence was associated with increased BOLD response in dorsomedial PFC, SMA, lateral orbitofrontal cortex (OFC), rectus, caudate, thalamus, amygdala, and MTG. Many of the activated areas such as the MTG, SMA [34], orbitofrontal cortex [46] caudate [47], and thalamus [49,50] suggest that the context required an increased emotional reprocessing (beyond the general context effect) when it made a previously positive picture negative.

In the field of emotion, activations in the lateral OFC in our study correspond to face selective part of OFC observed in a face discrimination reversal task when a formerly correct face was no longer the correct choice [59]. More specifically, in that study when a correct face was chosen, its expression turned into a smile, but when the wrong face was chosen, it turned into an angry face. In that study this part of OFC was activated when a formerly correct face was no longer the correct choice, so instead of a smile its choice resulted in an angry face. Thus, activation in this part of OFC was proposed to be error-related, as there was a discrepancy between the expected and perceived feedback. This area was recruited when face expression signaled a need for behavior change (i.e., change in the choices) [60]. On the basis of this, seeing a formerly positive face expression in an overall negative context–e.g., a smiling girl in a bullying context–also triggers error-related processes and might signal a need for behavior change. In our study, the biggest change in valence according to our post-test results emerged when positive pictures were put into a negative context. This change, or more precisely the shift from a positive to a negative meaning might require the reformulation of mental representations as well reflected in the increased activations in the supplementary motor area [34].

Medial PFC, especially its dorsomedial part has been activated in studies investigating empathy with other regions such as the SMA and thalamus [51]. In addition, medial prefrontal regions are recruited in reappraisal studies [61]. For instance, previous studies suggest that the regulation of negative emotions needs the allocation of cognitive resources provided by the dorsomedial prefrontal and dorsal cingulate gyrus [62]. The automatic shift from positive to negative in the meaning of the stimulus in our study might be accompanied by the recruitment of regions providing cognitive sources to regulate the resulting negative emotions. However, we did not find activation in the dorsolateral prefrontal cortex commonly observed in reappraisal studies [17,34,63].

We detected small activation in the amygdala that is often associated with negative/fearful emotional experiences, faces [64], emotional events and personal affective importance [65] or motivational relevance [66]. As this was the first fMRI study using the EST, we decided to present all significant activations. However, the cluster size was too small to interpret this result in the context of shifting.

The above results suggest that after excluding the general effect of context, shift in the meaning of a positive stimulus to negative induced by the context was supported in our task by areas involved in emotional processing, reformulation of mental representations, mentalizing, empathy and, to some extent, error-related process and cognitive control.

**Placing negative stimuli in positive context.** Interestingly, trials where a negative facial expression was placed into a positive context resulted in a less widespread activation pattern compared to trials where a positive facial expression turned out to have a negative meaning. The increased activation of MTG and fusiform gyrus suggested that this shift was also associated with increased emotional processing [67], whereas the recruitment of the precuneus also supports that affective and cognitive processing of affective mental states was also increased in the participants [51]. However, in this case, we did not detect activations in the medial or lateral part of the frontal cortex, suggesting that this automatic shift, or the resulting positive emotions, did not require extra cognitive resources or regulation at least in our task. Thus, the differences and similarities between effortful and automatic regulation of negative emotions require further studies.

Our results differ from those of Yang and colleagues [35]. Although they investigated spontaneous recovery from a negative emotional state as an implicit form of emotion regulation, we used context as a passive cue to trigger a change in emotional response, suggesting that different forms of automatic emotion regulation need to be tested in further studies.

**Placing negative stimuli in negative context and positive stimuli in positive context.** Interestingly, when a positive stimulus was placed into a positive context, or a negative stimulus was placed into a negative context, increased activation primarily in the occipital regions (calcarine, lingual gyrus and superior occipital lobule) was detected. In the literature, upregulating negative [68] or positive emotions [69] are mainly examined in reappraisal studies where participants are instructed to use certain tactics; thus, effortful and controlled processes are targeted. In our task, upregulation of emotions–confirmed by the valence and arousal ratings–was induced by the context that did not require effortful processes, which might explain our results. However, only small clusters were found in these two conditions after masking, so we should interpret these results with caution.

## Limitations

We asked participants to rate the valence and arousal of pictures during the post-task, not during the fMRI measure; however, results showed that the emotional valence ratings of the second, whole pictures changed in the expected direction. In addition, we did not record the eye

movements of the participants, so we cannot rule out the possibility that the four different conditions differed in the amount of eye movements, and that this may have affected our results.

In order to better understand the impact of the complex context in an emotional situation, adding conditions that include the context without the face might have been useful [70]. That would have allowed us to see if adding the face to the context would modify the valence beyond the information already evident in the context itself. However, the aim of the study was not to study facial and contextual information processing per se, but to use the context as a passive cue that promotes a shift in the meaning of certain set of images, therefore supporting emotion regulation.

Participants did not have to choose from emojis after the first picture, so we could not calculate reaction time differences within the pairs, but simply compared reaction times to second pictures, which limits our understanding; however, reaction time differences were not in the focus of our study. Additionally, it would have been ideal to put the same first picture in a negative and a positive context as well to directly compare how different contexts might influence the emotional processing of the same emotional stimulus.

Another limitation could be that participants might have had a certain expectation regarding the context that might have appeared in the activations to the second pictures. However, to avoid this or to decrease its possibility, participants were specifically instructed to solely focus on the stimuli on hand.

For the non-shift trials, stimuli were selected from the IAPS [37] database, whereas pictures for the shift trials were collected from the internet, so they are not from a standardized set of emotional stimuli; however, they went through several pilot studies [22]. Individual differences in the emotional reactivity, empathy, or ToM might affect the perception of emotional stimuli [71], but we did not assess these characteristics of our participants.

## Conclusion

We aimed to capture emotional flexibility by a task using context as a cue to trigger (an automatic) change in emotional response. The affective information and the social knowledge activated by the context had a major impact on the neural processing of the emotional visual stimuli. Presenting previously seen decontextualized emotional stimulus in a context recruited areas involved in emotional processing and understanding complex social situations, probably indicating that the context itself narrows the probability of emotions previously attributed to the decontextualized stimulus [72]. Thus, information about the context might be used as a cause of emotions [33].

Additionally, our results highlight that sensitivity and appropriate responses to context depend on many different processes; thus, emotional inflexibility may stem from different underlying mechanisms. Therefore, understanding emotional inflexibility in psychopathologies requires the dissection of these underlying mechanisms first.

## Supporting information

**S1 Fig. General context effect: Full pictures with the context vs. firstly presented images (pictures without the context) at p< .05, family-wise error (FWE) corrected for multiple comparison.** Coordinates are in Montreal Neurological Institute (MNI) space. Statistical maps were visualized on the MNI 152 template brain provided in MRIcroGL [39].
(DOCX)

**S1 Table. Valence and arousal values for the pictures of the post-task by contrasts.** P1: First picture of the picture pairs is positive. N1: First picture of the picture pairs is negative. P2:

Second picture of the picture pairs is positive. N2: Second picture of the picture pairs is nega-tive. + Results of the bootstrapped paired t-test. $^{++}$ Results of the Wilcoxon Signed Rank Test. $^{*}p < .001$. Valence and arousal were measured from 1 to 7 (1 being very unpleasant and 7 very pleasant; 1 being calm and 7 very excited, respectively).
(DOCX)

**S2 Table. Mean reaction times (in milliseconds) to the second picture in the scanner by the type of picture pairs in the Emotional Shifting Task.** P1: First picture of the picture pairs is positive. N1: First picture of the picture pairs is negative. P2: Second picture of the picture pairs is positive. N2: Second picture of the picture pairs is negative. Different letters (a, b, c) represent significant (p < .05) difference between mean scores, whereas the same letters repre-sent non-significant difference between mean scores according to the paired post hoc test of repeated measure of ANOVA.
(DOCX)

**S3 Table. General context effect: Increased activations to the 2nd pictures with context compared to the 1$^{st}$ pictures without context.** L = left; the initial statistical threshold was set to p< .05, family-wise error (FWE) corrected for multiple comparison.
(DOCX)

## Acknowledgments

The authors thank Mária Kelner for the drawing the first figure and Tamás Smahajcsik-Szabó for the figures with behavioural data.

## Author Contributions

**Conceptualization:** Brigitte Biró, Renáta Cserjési, Gabriella Juhász, Gyöngyi Kökönyei.

**Data curation:** Brigitte Biró, Gyöngyi Kökönyei.

**Formal analysis:** Brigitte Biró, Gyöngyi Kökönyei.

**Funding acquisition:** Gabriella Juhász.

**Investigation:** Brigitte Biró, Natália Kocsel, Attila Galambos, Kinga Gecse, Lilla Nóra Kovács, Dániel Baksa.

**Methodology:** Brigitte Biró, Renáta Cserjési, Natália Kocsel, Attila Galambos, Kinga Gecse, Dániel Baksa, Gabriella Juhász, Gyöngyi Kökönyei.

**Project administration:** Natália Kocsel, Attila Galambos, Lilla Nóra Kovács.

**Visualization:** Brigitte Biró, Gyöngyi Kökönyei.

**Writing – original draft:** Brigitte Biró, Gyöngyi Kökönyei.

**Writing – review & editing:** Brigitte Biró, Renáta Cserjési, Natália Kocsel, Attila Galambos, Kinga Gecse, Lilla Nóra Kovács, Dániel Baksa, Gabriella Juhász, Gyöngyi Kökönyei.

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
