## [Decision Letter · Decision Letter 0]

6 Sep 2022

PONE-D-22-11714The neural correlates of context driven changes in the emotional response: an fMRI studyPLOS ONE

Dear Dr. Kökönyei,

Thank you for submitting your manuscript to PLOS ONE. After careful consideration, we feel that it has merit but does not fully meet PLOS ONE’s publication criteria as it currently stands. Therefore, we invite you to submit a revised version of the manuscript that addresses the points raised during the review process.

When revising your manuscript, please in particular ensure you address the comments raised below regarding reporting details of the methodology and data.

We look forward to receiving your revised manuscript.

Kind regards,

Hugh Cowley

Staff Editor

PLOS ONE

2. "We note that Figure 1 includes an image of a [patient / participant / in the study].

6. We note that you have indicated that data from this study are available upon request. PLOS only allows data to be available upon request if there are legal or ethical restrictions on sharing data publicly. For more information on unacceptable data access restrictions, please see http://journals.plos.org/plosone/s/data-availability#loc-unacceptable-data-access-restrictions.

Reviewers' comments:

Reviewer's Responses to Questions

**Comments to the Author**

1. Is the manuscript technically sound, and do the data support the conclusions?

Reviewer #1: Yes

Reviewer #2: Yes

Reviewer #3: Partly

2. Has the statistical analysis been performed appropriately and rigorously? 

Reviewer #1: Yes

Reviewer #2: Yes

Reviewer #3: Yes

3. Have the authors made all data underlying the findings in their manuscript fully available?

Reviewer #1: Yes

Reviewer #2: Yes

Reviewer #3: Yes

4. Is the manuscript presented in an intelligible fashion and written in standard English?

Reviewer #1: Yes

Reviewer #2: No

Reviewer #3: No

5. Review Comments to the Author

Reviewer #1: OVERALL IMPRESSION

The authors have investigated the neural correlates of emotional flexibility. The paper is innovative with its two focal points combined in one study: the effect of context and the direction of emotion regulation.

Below, please find some suggestions to improve the manuscript.

SPECIFIC COMMENTS

ABSTRACT

1. As the direction of emotion regulation is a key aspect in the study (besides the effect of context), I recommend to clarify it in the abstract (before line 36, in which a ‘shift from positive to negative valence’ is mentioned).

2. It is unclear in the abstract (but clear in the main text) why facial emotional processing and responsible brain regions are relevant. (Because the decontextualized details of the pictures are faces.) It would be great to clarify it in the abstract.

3. Brain structures are identified while the authors report the results in the abstract, except for line 34 where funtions are mentioned (‘areas involved in facial emotional processing and affective mentalizing’). Some important regions could be added here.

INTRODUCTION

4. Why is it an upregulation to have a positive first picture followed by a positive second picture? Does the valence rating increase (i.e., valence for the second picture with context > valence for the first picture without context? It seems to be ‘only’ a consistency between first and second picture compared to the inconsist trials (negative first picture – positive second picture and vice verse). (see from line 122 - Table 1). Of course, the Results part gives an answer to the question above, but there is a slight inconsistency in using terms in different parts of the manuscript, such as upregulation, shifting, context effect. Please, correct it.

METHODS

5. Lines 156 – 159: Although, I understand that Reference #29 contains the methodological background of EST, it would be useful to add some more information about the stimulus material (e.g., database, ID number of selected pictures; selection criteria for depicting social scenes/faces/emotional expression etc).

6. Was the smaller detail of the pairs of picture (contextualized) always a face/facial expression? If yes, it would be great to clarify it in the text. (lines 156-159).

7. Line 159: What was the purpose of the happy/sad emojis? Please, add a short explanation.

RESULTS

8. Lines 193-194 and 241-242: Please, check the valence ratings in the non-shift condition (S1 Table: 5.850 vs 6.129), if there is a significant difference between them. (This difference seems to be too small.)

9. Lines 251-252 and S1 Table: The changes in valence ratings are in the focus of the study, however, it is very interesting that arousal ratings are always higher for the second pictures than for the first ones. What could be the reason for that? Do you have an explanation?

DISCUSSION

10. Lines 338-342: Please, explain the top-down manner of changes in valence and arousal (due to the context) more detailed.

11. Lines 472-473: It is unclear, what is the exact limitation related to the pictures (from IAPS database and the collected pictures from the internet)?

IN SUM

This is a well-designed, innovative fMRI-study to explore the effect of context and valence. I recommend it for acceptance after minor changes.

Reviewer #2: In this paper, the Authors aim to study the mechanisms of emotional flexibility (i.e., the change of emotional responses according to the context), performing both behavioral tests (valence, arousal, reaction times) and functional brain imaging (fMRI).

Major comments:

Line 359: Eye movements should be taken in consideration as a possible explanation of occipital activity; the Authors should mention the lack of eye movement recording among the limitations of the study

"facial expression" appears quite late in the manuscript; it should be mentioned in Keywords and Abstract (and/or title?)

The mask used in the analysis is mentioned in different manners: e.g., line 35 "explicit", line 131 "external", line 273 "exclusive"; the Authors should use consistently the most appropriate one and explain clearly what it means

Table 1 not necessary, same concept is adequately explained in text

Line 443: it should probably read "context"?

Line 445: is it referred to "negative to negative", not " negative to positive"?

Minor points:

line 147: "standardized handedness questionnaire": commonly quoted as "Edimburgh inventory"

Several language imperfections, too many to mention them all (the Authors should have the English checked throughout); some examples:

line 38 ans 138: it should read "it resulted IN a"

line 61: it should read: "pride instead of sadness"

line 128: "/even": ?

line 146 and 166: parenthesis missing

line 151: "and were excluded with any history...": grammar!?

line 187: "crosses were presented with variable duration"

line 199: "Four hundred and nine volumes were acquired...."

line 372: "inferring affective states OF others"

line 376 and following: "Beyond its role ... reported arousal": grammar!?

Reviewer #3: Biró et al., explored the neural correlates of emotional response during context shifting. This work links emotional context shift to changes in the BOLD response. This work may be of interest to communities investigating the role of context during emotional processing. I have comments and questions about the manuscript that need clarification.

1. I recommend that the authors review the grammar of the manuscript. Some errors disrupt the clarity of writing, e.g., hyphens (line 57, 118, 119, 415) and parentheses (line 88, 411, 430, 440) when not necessary.

2. Further, the authors should address the structure of the manuscript. There are several one-sentence paragraphs throughout the manuscript, e.g., lines 203, 241, 364, 393, 427, and 453. The flow of the introduction and discussion should be improved to paint a clearer picture of the existing literature and interpretation of the results.

3. On lines 245/256 the authors discuss the accuracy of participants' responses. The values in parentheses (mean: 22.93, range 19-24) do not seem to reflect the accuracy rate which leads me to believe they are an error, could the authors please explain these values?

4. In the methods section of the manuscript, the authors detail how behavioural data will be analysed with “descriptive and non-parametric statistics”. I suggest the authors use non-parametric tests where the assumptions for parametric tests have been violated and specify this in-text. Further, I assume non-parametric tests have been employed due to violation of normality, have the authors considered transforming the data to better fit the normal distribution?

5. It would be beneficial to the reader to specify the levels of the ANOVA in the results section.

6. In the supplementary material, table S2 specifies in the legend: “Different letters (a, b, c) represent significant (p < .05) difference between mean scores, whereas the same letters represent non-significant difference between mean scores according to the paired post hoc test of repeated measure of ANOVA". I find this difficult to follow and suggest the authors present all behavioural results as figures in-text, with significance denoted by asterisks.

7. When using acronyms, use the full term first followed by the acronym in parentheses, e.g., Theory of Mind (ToM).

8. There are details of stimuli selection from the International Affective Picture System (IAPS) in the discussion section, which is not included in the methods section of the manuscript. Please ensure any relevant information about the stimuli and/or task is included in the methods section.

9. The authors should ensure they use the same referencing style throughout the manuscript, e.g., lines 397- 399.

10. There are several small clusters after masking, e.g., cluster size = 2 for amygdala activation for positive pictures in a negative context > positive pictures without context. How can the authors be sure they are not over-interpreting small clusters?

6. PLOS authors have the option to publish the peer review history of their article (what does this mean?). If published, this will include your full peer review and any attached files.

Reviewer #1: No

Reviewer #2: No

Reviewer #3: **Yes: **Jessica Henderson

---

## [Author Response · Author response to Decision Letter 0]

16 Nov 2022

Dear Editor-in-Chief, Dear Dr. Hugh Cowley, 

We would like to thank you for giving us the opportunity to improve our manuscript. The comments and suggestions of the reviewers helped us to structure the Introduction part in a more logic way. Questions on the task and results helped us to provide a clearer overview of our study, and to complete the discussion with further relevant thoughts. As a result, we would like to take the opportunity and resubmit the revised version of our manuscript, entitled “The neural correlates of context driven changes in the emotional response: an fMRI study”.

Please, find below our detailed answers to the reviewer’s comments (written in blue in the response to reviewers.doc). In the revised manuscript the original text is presented in black, modifications are indicated by tracked changes mode in MS Word. 

We used a professional English language editing service that substantially helped to increase the readability of the manuscript. Please note that these modifications are not marked by tracked changes in the manuscript.

We were asked to address the following additional requirements:

1. We checked all the PLOS ONE's style requirements, and manuscript meets the requirements, as does the naming of the files. 

2. We would like to note that Figure 1 did not include an image of a patient/participant in the study. Instead, Figure 1 contained pictures from the International Affective Picture System and pictures from the internet which were collected for the shifting of trials. However, in order to avoid copyright issues, we have decided to restructure Figure 1 and present our task schematically. The revised manuscript contains the new schematic presentation of the Emotional Shifting Task. 

3. Grant information we provided in the ‘Funding Information’ and ‘Financial Disclosure’ sections match.

Funding

This study was supported by the Hungarian Academy of Sciences (MTA-SE Neuropsychopharmacology and Neurochemistry Research Group), the Hungarian Brain Research Program (Grant: 2017-1.2.1-NKP-2017-00002), and the Hungarian Brain Research Program 3.0 (NAP2022-I-4/2022), and by the Hungarian National Research, Development and Innovation Office (Grant No. FK128614, K 143391). Project no. TKP2021-EGA-25 has been implemented with the support provided by the Ministry of Innovation and Technology of Hungary from the Hungarian National Research, Development and Innovation Fund, financed under the TKP2021-EGA funding scheme. DB was supported by the ÚNKP-20-3-II-SE-51 New National Excellence Program of the Ministry for Innovation and Technology from the source of the National Research, Development and Innovation Fund. The sponsors had no role in the design of study, in the collection, analysis, interpretation of data and in the writing the manuscript.

4. In order to use the direct billing option I the corresponding author amended her affiliation with the chosen institute (Semmelweis University)

5. Regarding data availability statement, we have specified that data regarding valence, arousal ratings and reaction times, along with fMRI contrast maps are fully available at the following link: https://osf.io/hgdky/. However, we are not allowed to share raw imaging dataset publicly, because at the time our study started, there was no information on open access data availability in the consent forms (the study was approved by the Scientific and Research Ethics Committee of the Medical Research Council (Hungary)), therefore participants were not able to accept or refuse their assent to share imaging data in an open access repository. However, raw imaging data are available from the corresponding author (Gyöngyi Kökönyei, kokonyei.gyongyi@ppk.elte.hu) or from the Department of Pharmacodynamics, Faculty of Pharmacy, Semmelweis University (titkarsag.gyhat@pharma.semmelweis-univ.hu) on reasonable request.

Should any other issues with the manuscript arise, please let us know and we are prepared to address them. 

Sincerely,

the authors

Reviewers' comments:

Reviewer #1: OVERALL IMPRESSION

The authors have investigated the neural correlates of emotional flexibility. The paper is innovative with its two focal points combined in one study: the effect of context and the direction of emotion regulation.

Answer# Thank you for your positive feedback and for taking the time to review our manuscript!

Below, please find some suggestions to improve the manuscript.

SPECIFIC COMMENTS

ABSTRACT

1. As the direction of emotion regulation is a key aspect in the study (besides the effect of context), I recommend to clarify it in the abstract (before line 36, in which a ‘shift from positive to negative valence’ is mentioned).

Answer# Thank you for your comment. Since emotion regulation and its direction are key aspects in the study, these should be definitely included in the abstract. 

In the text: To understand how context can trigger a change in emotional response, i.e. how it can up-regulate the initial emotional response or trigger a shift in the valence of emotional response, we used a task consisting of picture pairs during functional magnetic resonance imaging sessions.

2. It is unclear in the abstract (but clear in the main text) why facial emotional processing and responsible brain regions are relevant. (Because the decontextualized details of the pictures are faces.) It would be great to clarify it in the abstract.

Answer# We added this information to the abstract.

In the text: In each pair the first picture was a smaller detail (a decontextualized photograph depicting emotions using primarily facial and postural expressions) from the second (contextualized) picture and the neural response to a decontextualized picture was compared with the same picture in a context.

3. Brain structures are identified while the authors report the results in the abstract, except for line 34 where functions are mentioned (‘areas involved in facial emotional processing and affective mentalizing’). Some important regions could be added here.

Answer# Thank you for your suggestions. We added some relevant regions to facial emotional processing and affective mentalizing.

In the text: In general, context (vs. pictures without context) increased activation in areas involved in facial emotional processing (e.g., middle temporal gyrus, fusiform gyrus, and temporal pole) and affective mentalizing (e.g., precuneus, temporoparietal junction). 

INTRODUCTION

4. Why is it an upregulation to have a positive first picture followed by a positive second picture? Does the valence rating increase (i.e., valence for the second picture with context > valence for the first picture without context (It seems to be ‘only’ a consistency between first and second picture compared to the inconsistant trials (negative first picture – positive second picture and vice versa). (see from line 122 - Table 1). 

Answer# We used term upregulation in cases where the valence of the same initial emotional response increased. We tested the changes in the valence and arousal after scan. Indeed, when a positive first picture was followed by a positive second picture, both the valence and the arousal post-scan ratings increased. In S1 Table we presented all the data relevant to these comparisons. Since Reviewer 3 suggested to use figures to help readers to understand supplementary tables and changes in valence and arousal induced by the context, separate figures (Fig 3 and 4) show the valence and arousal data for the shift and upregulation conditions (see below). The figure shows when a first positive picture followed by second positive picture (P1P2 condition), valence became more positive. Similarly, when a first negative picture followed by a second negative picture (N1N2 condition), the valence became more negative. These changes were significant. In addition, changes in arousal are also significant in both P1P2 and N1N2 conditions.

Fig 3. Changes in valence ratings in the post-task. Note. P1: First picture of the picture pairs is positive. N1: First picture of the picture pairs is negative. P2: Second picture of the picture pairs is positive. N2: Second picture of the picture pairs is negative. * p < .001.

Fig 4. Changes in arousal ratings in the post-task. Note. P1: First picture of the picture pairs is positive. N1: First picture of the picture pairs is negative. P2: Second picture of the picture pairs is positive. N2: Second picture of the picture pairs is negative. * p < .001

Of course, the Results part gives an answer to the question above, but there is a slight inconsistency in using terms in different parts of the manuscript, such as upregulation, shifting, context effect. Please, correct it.

Answer# Thank you for your comment. Indeed, we use all three terms in the text: upregulation, shift and context effect. However, the three terms are used to describe completely different phenomena. The term upregulation is used in cases where the valence of the same initial emotional response increases, so that when a negative image is presented in a negative context, or a positive image is presented in a positive context. In these cases the initial negative response becomes more negative and the initial positive response becomes more positive. By shift we refer to the process where the valence of the initial emotional response is reversed, i.e., the initial negative emotional response becomes positive and vice versa. The term general context effect, on the other hand, is used in a general sense, i.e., it refers only to the process of responding to an emotional stimulus in a context vs. responding to the same stimulus without context.

Based on you comment, these three terms have been clarified in the introduction to make it easier to follow which process is being discussed. We also checked that the correct one of the three terms was used in any part of the manuscript.

In the text: By shift we refer to categorical changes where the valence of the initial emotional response is reversed, i.e., the initial negative emotional response becomes positive and vice versa. The term upregulation is used when the valence of the same initial emotional response is increased by the context; thus, a negative stimulus becomes more negative, or a positive one becomes more positive. Accordingly, the EST contained two shift and two non-shift (upregulation) conditions.

On the basis of the theory by Saxe and Houlihan [33] different emotional responses could be expected to stimuli in context vs. without context. They argue that forward inferences are used to attribute emotions to the target when an emotional expression is processed in a context; thus, we automatically infer that the cause of the emotional state of the target reflected in their emotional expressive behavior is the context /event. On the basis of this, we expected that context itself would recruit areas involved in emotional processing and understanding complex social situations; thus, first we simply compared the neural responses to whole pictures vs. decontextualized (cropped) pictures. We refer to this as a general context effect in our study.

METHODS

5. Lines 156 – 159: Although, I understand that Reference #29 contains the methodological background of EST, it would be useful to add some more information about the stimulus material (e.g., database, ID number of selected pictures; selection criteria for depicting social scenes/faces/emotional expression etc).

Answer# As you suggested we added more information about the stimulus material to the Method section. We would like to mention that in order to avoid copyright issues, we have decided to restructure Figure 1 and present our task schematically. The revised manuscript contains the new schematic presentation of the Emotional Shifting Task.

In the text: For the upregulation conditions (P1P2 and N1N2), pictures were selected from the International Affective Picture System [37]. Their identification numbers were 1340, 2091, 2141, 2205, 2216, 2340, 2530, 2700, 6242, 6838, 8497, and 9050. For the shift conditions (P1N2 and N1P2) pictures were selected from the internet. Six criteria were used to select the images: (1) free for non-commercial use, (2) depicting social interactions, (3) evoking an emotional response without being shocking or extreme, (4) not depicting famous person(s), (5) eligible for shifting conditions, i.e., the valence of facial expression and the whole picture should be opposite, and (6) the images should represent as many different situations as possible.

6. Was the smaller detail of the pairs of picture (contextualized) always a face/facial expression? If yes, it would be great to clarify it in the text. (lines 156-159). 

Answer# Not always, but in most cases the cropped image expressed emotion primarily through facial expression and/or posture. We added this information to the text: 

In the text: The EST [22] consists of 24 picture pairs. In each pair, the first picture is always a smaller detail from the second (whole) picture. In most cases the cropped image expressed emotion primarily through facial expression and/or posture. The valence of the firstly presented picture either remains or changes when it is placed into a context, and so should change the elicited emotion (Fig 1).

7. Line 159: What was the purpose of the happy/sad emojis? Please, add a short explanation. 

Answer# Thank you for your question. We decided to use happy/sad emojis in the scanner to mimic the two endpoints of valence ratings of Self-Assessment Manikin. This was an easy and quick method to check the valence of second (whole) picture while performing the task. 

In the text: After each pair, a happy and a sad smiley/emoji was shown on the screen (Fig 2), and participants had to choose one of them by pressing the corresponding button to indicate the valence (positive or negative) of the second (whole) picture. We decided to use emojis in the scanner to mimic the two endpoints of valence ratings in Self-Assessment Manikin (Lang et al., 1997).

RESULTS

8. Lines 193-194 and 241-242: Please, check the valence ratings in the non-shift condition (S1 Table: 5.850 vs 6.129), if there is a significant difference between them. (This difference seems to be too small.)

Answer# Thank you for your comment. We have checked all the analyses, and all the differences were significant. We also repeated our analyses with bootstrapped paired t-tests, as it is easier to interpret changes in means than in ranks. We completed S1 table with these t-tests results. 

9. Lines 251-252 and S1 Table: The changes in valence ratings are in the focus of the study, however, it is very interesting that arousal ratings are always higher for the second pictures than for the first ones. What could be the reason for that? Do you have an explanation?

Answer# Thank you for your question. Based on the paper by Saxe and Houlihan (2017), our idea is that the context might provide extra affective information, since it gives an explanation for the emotion; thus, it guides our interpretation. For instance, seeing a crying woman in hospital compared to seeing just a crying woman, could elicit a more intense emotional response, as it can activate additional emotional meaning or knowledge such as her relative being sick. 

In the text: Significant changes in the valence and arousal ratings of the stimuli were observed after they were placed into a context, indicating that participants reinterpreted the emotional stimuli. These changes in the valence and arousal were detected in all four types of picture sets, indicating that the context itself did not only shape the categorization of the emotional states [9], but it also might affect valence and intensity (arousal) in a top-down manner. The context, or more precisely the appraisal of overall context (e.g., bullying, being in a hospital with a sick person, or childbirth), i.e., the semantic features, might provide extra affective information, as it gives an explanation for the emotion; thus, it guides our interpretation [33]. For instance, seeing a crying woman in hospital compared to seeing just a crying woman could elicit a more intense emotional response, as it can activate additional emotional meaning or knowledge such as her relative being sick. 

DISCUSSION

10. Lines 338-342: Please, explain the top-down manner of changes in valence and arousal (due to the context) more detailed. 

Answer# Thank you for your question. We used the term top-down manner because we believed that the appraisal/meaning of overall context (e.g. bullying, being in a hospital with a sick person, or childbirth), i.e., the semantic features, guided the changes. We included this idea above (see our answer to question 9). However, it does not necessarily mean that it happened in a top-down manner, since declarative knowledge and experiences could be retrieved automatically and actively (Osada et al., 2008, Philos Trans R Soc Lond B Biol Sci, 363(1500):2187-99.). In addition, we can’t exclude that that low level image features are not relevant in the observed changes. Therefore, we decided to delete “in a top-down manner” from the text. 

11. Lines 472-473: It is unclear, what is the exact limitation related to the pictures (from IAPS database and the collected pictures from the internet)? 

Answer# We wanted to mention that pictures in the shift trials are not from a standardized set of stimuli, however we piloted the pictures before the fMRI study. 

In the text: For the non-shift trials, stimuli were selected from the IAPS [37] database, whereas pictures for the shift trials were collected from the internet, so they are not from a standardized set of emotional stimuli; however, they went through several pilot studies [22].

This is a well-designed, innovative fMRI-study to explore the effect of context and valence. I recommend it for acceptance after minor changes.

Answer# Thank you very much for your positive and encouraging feedback, for your comments, suggestions and questions. They helped us a lot in improving our manuscript. 

Reviewer #2: In this paper, the Authors aim to study the mechanisms of emotional flexibility (i.e., the change of emotional responses according to the context), performing both behavioral tests (valence, arousal, reaction times) and functional brain imaging (fMRI).

Major comments:

Line 359: Eye movements should be taken in consideration as a possible explanation of occipital activity; the Authors should mention the lack of eye movement recording among the limitations of the study 

Answer # Thank you for bringing this to our attention. We added the lack of eye movement recording to the section of Limitations. 

In the text: In addition, we did not record the eye movements of the participants, so we cannot rule out the possibility that the four different conditions differed in the amount of eye movements, and that this may have affected our results.

"facial expression" appears quite late in the manuscript; it should be mentioned in Keywords and Abstract (and/or title?)

Answer# Thank you for your suggestion. We added this information to the abstract and “facial expression” to the Keywords.

In the text: In each pair, the first picture was a smaller detail (a decontextualized photograph depicting emotions using primarily facial and postural expressions) from the second (contextualized) picture, and the neural response to a decontextualized picture was compared with the same picture in a context.

Keywords: emotional flexibility; context; neuroimaging; emotion processing; mentalization; empathy; facial expression;

The mask used in the analysis is mentioned in different manners: e.g., line 35 "explicit", line 131 "external", line 273 "exclusive"; the Authors should use consistently the most appropriate one and explain clearly what it means

Answer# Thank you for your comment. Indeed, we mentioned the mask used in the analysis in different manners in the manuscript. We decided to use the term exclusive mask, as we wanted to exclude from the analysis the activation observed for the general context effect. We have corrected each word in the text to this term everywhere. 

For instance, in the text: On the basis of this, we expected that context itself would recruit areas involved in emotional processing and understanding complex social situations: thus, first we simply compared the neural responses to whole pictures vs. decontextualized (cropped) pictures. We refer to this as a general context effect in our study. Then we used this activation map as an external exclusive mask to be able to explore the four different types of automatic changes specifically in emotional responses. It allowed us to explore neural activation to changes in the meaning triggered by the context as a passive cue independent of the context vs. no context differences.

Table 1 not necessary, same concept is adequately explained in text

Answer# Thank you for your comment. Indeed, we aimed to explain the main concepts in the text, but following the suggestion of Reviewer 1, we decided to keep Table 1 as well, to clarify the terms used in the manuscript.

We would like to mention that in order to avoid copyright issues, we have decided to restructure Figure 1 and present our task schematically. The revised manuscript contains the new schematic presentation of the Emotional Shifting Task.

Line 443: it should probably read "context"?

AnswerXYes, of course. Thank you, we corrected it accordingly.

In the text: Placing negative stimuli in negative context and positive stimuli in positive context

Line 445: is it referred to "negative to negative", not " negative to positive"?

Answer# Thank you. We corrected it accordingly.

In the text: Interestingly, when a positive stimulus was placed into a positive context, or a negative stimulus was placed into a negative context increased activation primarily in the occipital regions (calcarine, lingual gyrus and superior occipital lobule, respectively) was detected.

Minor points:

line 147: "standardized handedness questionnaire": commonly quoted as "Edinburgh inventory"

Answer# We added the name of the inventory to the text.

In the text: The participants were right-handed, as assessed by the Edinburgh Handedness Inventory [35], and had normal or corrected-to-normal vision.

Several language imperfections, too many to mention them all (the Authors should have the English checked throughout); some examples:

line 38 ans 138: it should read "it resulted IN a"

line 61: it should read: "pride instead of sadness"

line 128: "/even": ?

line 146 and 166: parenthesis missing

line 151: "and were excluded with any history...": grammar!?

line 187: "crosses were presented with variable duration"

line 199: "Four hundred and nine volumes were acquired...."

line 372: "inferring affective states OF others"

line 376 and following: "Beyond its role ... reported arousal": grammar!?

Answer# Thank you for this suggestion, we used a professional English language editing service that substantially helped to increase the readability of the manuscript. Please note that these modifications are not marked by tracked changes in the manuscript.

Thank you very much for your critics, comments and questions. They helped us improve our manuscript. We hope that we have managed to address your concerns.

Reviewer #3: Biró et al., explored the neural correlates of emotional response during context shifting. This work links emotional context shift to changes in the BOLD response. This work may be of interest to communities investigating the role of context during emotional processing. I have comments and questions about the manuscript that need clarification.

1. I recommend that the authors review the grammar of the manuscript. Some errors disrupt the clarity of writing, e.g., hyphens (line 57, 118, 119, 415) and parentheses (line 88, 411, 430, 440) when not necessary.

Answer# Thank you for this suggestion, we used a professional English language editing service that substantially helped to increase the readability of the manuscript. Please note that these modifications are not marked by tracked changes in the manuscript.

2. Further, the authors should address the structure of the manuscript. There are several one-sentence paragraphs throughout the manuscript, e.g., lines 203, 241, 364, 393, 427, and 453. 

Answer# Thank you for the comment. Indeed there are some one-sentence paragraphs in the Method and Discussion sections. In the Method section we added additional sentences to these paragraphs since you asked us to provide some further details about the analyses of behavioral data (e.g. about non-parametric/parametric analyses). We also checked the discussion to complete paragraphs with one sentence.

The flow of the introduction and discussion should be improved to paint a clearer picture of the existing literature and interpretation of the results.

Answer# Thank you for your feedback. We restructured the introduction to show why the effect of context on the emotional processing may be relevant for understanding emotional flexibility. Thus, we first introduced the concept of emotional flexibility highlighting the role of adjustment of our emotional response to the context. Next, we argued that emotion perception is context dependent, and then we cited cognitive reappraisal studies to support the idea that creating a new cognitive context for an emotional stimulus has relevance for understanding context effect on emotional trajectories. Next, we introduced the EST task, and we wrote about how this task can capture shift and upregulation of emotional responses. And finally, we presented our expectation on neural activity based on the literature. 

Regarding the discussion, we have kept the original order of the topics, but we have added a number of points to the text to make the argument more fluid. 

3. On lines 245/256 the authors discuss the accuracy of participants' responses. The values in parentheses (mean: 22.93, range 19-24) do not seem to reflect the accuracy rate which leads me to believe they are an error, could the authors please explain these values?

Answer# Thank you! Of course, the data in the brackets were wrong. We corrected it.

In the text: This accuracy was 95.54% (range: 79-100%).

4. In the methods section of the manuscript, the authors detail how behavioural data will be analysed with “descriptive and non-parametric statistics”. I suggest the authors use non-parametric tests where the assumptions for parametric tests have been violated and specify this in-text. Further, I assume non-parametric tests have been employed due to violation of normality, have the authors considered transforming the data to better fit the normal distribution?

Answer# When analysing the change in valence and arousal we used non-parametric tests due to violation of normality. However, as it is easier to interpret changes in means than in ranks, we repeated these analyses using bootstrapped paired t-tests. In S1 Table means are reported instead of ranks. Thus, instead of transforming the data to better fit the normal distribution we preferred to use bootstrapped paired t-test, since bootstrapping improve the power of t-test under violation of non-normality (e.g. Konietschke & Pauly, 2013). 

We added these pieces of information to the section of Statistical analysis of self-report and post-test data and completed S1 Table with changes in means and the corresponding 95% confidence intervals. 

In the text: Since the distribution of valence and arousal ratings was non-normal, we used Wilcoxon Signed Rank Test to compare the valence and arousal ratings of the first and second pictures in each condition (P1N2, N1P2, P1P2, and N1N2). However, as it was easier to interpret changes in means than in ranks, we repeated these analyses using a series of bootstrapped paired t-tests. A repeated measures ANOVA was performed on the reaction times collected during the fMRI scan.

5. It would be beneficial to the reader to specify the levels of the ANOVA in the results section.

Answer# We used a repeated measures ANOVA on the reaction times collected during the fMRI scan, thus we compared the reaction times in the four conditions (P1N2, P1P2, N1P2, N1N2). To make it clearer, we have added this information to the section of Statistical analysis of self-report and post-test data.

In the text: A repeated measures ANOVA was performed on the reaction times collected during the fMRI scan.

6. In the supplementary material, table S2 specifies in the legend: “Different letters (a, b, c) represent significant (p < .05) difference between mean scores, whereas the same letters represent non-significant difference between mean scores according to the paired post hoc test of repeated measure of ANOVA". I find this difficult to follow and suggest the authors present all behavioural results as figures in-text, with significance denoted by asterisks.

Answer # As you suggested, we present all the behavioral results (including results on arousal, valence and reaction times) as figures in text (Fig 3-5), while tables with behavioral results are supplementary ones. Fig 3. and Fig 4. 

Fig 5. Mean reaction times (and standard deviations in milliseconds) to the second picture of the picture pairs in the scanner by the type of picture pairs in the Emotional Shifting Task. Note. P1: First picture of the picture pairs is positive. N1: First picture of the picture pairs is negative. P2: Second picture of the picture pairs is positive. N2: Second picture of the picture pairs is negative. * p < .05

7. When using acronyms, use the full term first followed by the acronym in parentheses, e.g., Theory of Mind (ToM).

Answer# We checked all the acronyms in the text. 

8. There are details of stimuli selection from the International Affective Picture System (IAPS) in the discussion section, which is not included in the methods section of the manuscript. Please ensure any relevant information about the stimuli and/or task is included in the methods section.

Answer# Thank you for your comment. We completed the Methods section with the relevant information. We would like to mention that in order to avoid copyright issues, we have decided to restructure Figure 1 and present our task schematically. The revised manuscript contains the new schematic presentation of the Emotional Shifting Task.

In the text: For the upregulation conditions (P1P2 and N1N2), pictures were selected from the International Affective Picture System [37]. Their identification numbers were 1340, 2091, 2141, 2205, 2216, 2340, 2530, 2700, 6242, 6838, 8497, and 9050. For the shift conditions (P1N2 and N1P2) pictures were selected from the internet. Six criteria were used to select the images: (1) free for non-commercial use, (2) depicting social interactions, (3) evoking an emotional response without being shocking or extreme, (4) not depicting famous person(s), (5) eligible for shifting conditions, i.e., the valence of facial expression and the whole picture should be opposite, and (6) the images should represent as many different situations as possible.

9. The authors should ensure they use the same referencing style throughout the manuscript, e.g., lines 397- 399.

Answer# Thank you. We corrected the text accordingly. 

In the text: Many of the activated areas such as the MTG, SMA [34], orbitofrontal cortex [46] caudate [47], and thalamus [49, 50] suggest that the context required an increased emotional reprocessing (beyond the general context effect) when it made a previously positive picture negative.

10. There are several small clusters after masking, e.g., cluster size = 2 for amygdala activation for positive pictures in a negative context > positive pictures without context. How can the authors be sure they are not over-interpreting small clusters?

Answer# We were hesitating whether to present such small clusters. As this was the first test of the task, we finally added them to the tables since we set the threshold to p< .05, family-wise error (FWE) corrected for multiple comparison. However, we agree that we should avoid overinterpreting our results, so we deleted amygdala from the abstract and added some sentences to the discussion. We also have to admit that we found small clusters in the upregulation conditions after masking, so we should interpret those results with caution. 

In the text: We detected small activation in the amygdala that is often associated with negative/fearful emotional experiences, faces [64], emotional events and personal affective importance [65] or motivational relevance [66]. As this was the first fMRI study using the EST, we decided to present all significant activations. However, the cluster size was too small to interpret this result in the context of shifting. 

In the text: However, only small clusters were found in these two conditions after masking, so we should interpret these results with caution. 

Answer# Thank you very much for your critics, comments and questions, they helped us improve our manuscript. We hope that we have managed to address your concerns.

---

## [Editor Report · Decision Letter 1]

15 Dec 2022

The neural correlates of context driven changes in the emotional response: an fMRI study

PONE-D-22-11714R1

Dear Dr. Kökönyei,

We’re pleased to inform you that your manuscript has been judged scientifically suitable for publication and will be formally accepted for publication once it meets all outstanding technical requirements.

Kind regards,

Fausta Lui

Guest Editor

PLOS ONE
---

## [Editor Report · Acceptance letter]

21 Dec 2022

PONE-D-22-11714R1 

The neural correlates of context driven changes in the emotional response: an fMRI study 

Dear Dr. Kökönyei:

I'm pleased to inform you that your manuscript has been deemed suitable for publication in PLOS ONE. Congratulations! Your manuscript is now with our production department. 

Kind regards, 

on behalf of

Dr. Fausta Lui 

Guest Editor

PLOS ONE